# LaTo: Landmark-tokenized Diffusion Transformer for Fine-grained Human Face Editing

**Zhenghao Zhang**[1]* **Ziying Zhang**[2]* **Junchao Liao**[1]* **Xiangyu Meng**[1]
**Qiang Hu**[2] **Siyu Zhu**[3] **Xiaoyun Zhang**[2✉] **Long Qin**[1✉] **Weizhi Wang**[1]
[1] Alibaba Cloud Computing [2] Shanghai Jiao Tong University [3] Fudan University

## Abstract

Recent multimodal models for instruction-based face editing enable semantic manipulation but still struggle with precise attribute control and identity preservation. Structural facial representations such as landmarks are effective for intermediate supervision, yet most existing methods treat them as rigid geometric constraints, which can degrade identity when conditional landmarks deviate significantly from the source (e.g., large expression or pose changes, inaccurate landmark estimates). To address these limitations, we propose LaTo, a landmark-tokenized diffusion transformer for fine-grained, identity-preserving face editing. Our key innovations include: (1) a landmark tokenizer that directly quantizes raw landmark coordinates into discrete facial tokens, obviating the need for dense pixel-wise correspondence; (2) a location-mapped positional encoding and a landmark-aware classifier-free guidance that jointly facilitate flexible yet decoupled interactions among instruction, geometry, and appearance, enabling strong identity preservation; and (3) a landmark predictor that leverages vision–language models to infer target landmarks from instructions and source images, whose structured chain-of-thought improves estimation accuracy and interactive control. To mitigate data scarcity, we curate HFL-150K, to our knowledge the largest benchmark for this task, containing over 150K real face pairs with fine-grained instructions. Extensive experiments show that LaTo outperforms state-of-the-art methods by 7.8% in identity preservation and 4.6% in semantic consistency. Code is available at https://github.com/alibaba/landmark-tokenized-dit .

## 1 Introduction

Generating photorealistic facial images with controllable expression, pose, and attributes while preserving identity remains a central challenge in face editing (Lin et al., 2025; Preechakul et al., 2022; Zhang et al., 2025). This capability is essential for applications such as virtual avatars, digital humans, and identity-preserving face manipulation. Recent proprietary multimodal models, including SeedEdit3 (Wang et al., 2025), Step1X-Edit (Liu et al., 2025), and FLUX.1-Kontext (Labs et al., 2025), have significantly improved instruction-based editing by leveraging large-scale vision-language modeling (Li et al., 2024; Deng et al., 2025). Yet fine-grained control still requires detailed, standardized prompts, and existing models often struggle with accurate instruction following and identity preservation during in-context generation (Liu et al., 2025; Xiao et al., 2025; Labs et al., 2025). We attribute these failures to their reliance on high-level semantic encoders, which inadequately capture the structural cues needed for precise facial control.

Facial landmarks are a common structural prior for improving edit fidelity because they provide explicit 2D geometric constraints over key facial features, localizing edits more precisely than text prompts (Yang & Guo, 2020; Wei et al., 2025; Liang et al., 2024; Li et al., 2022; Sun et al., 2024). However, most landmark-based methods are built on GANs or UNet-based diffusion models (Goodfellow et al., 2020; Ronneberger et al., 2015) and transfer poorly to modern DiTs (Peebles & Xie, 2023). Recent DiT-based editors such as OminiControl (Tan et al., 2024) and OmniGen (Xiao et al., 2025) rasterize landmarks into 2D images, encode them with a VAE (Kingma & Welling, 2022), and use the resulting dense visual tokens as guidance. While effective, this design encourages pixel-level

---

*These authors contributed equally to this work. ✉ Corresponding authors.

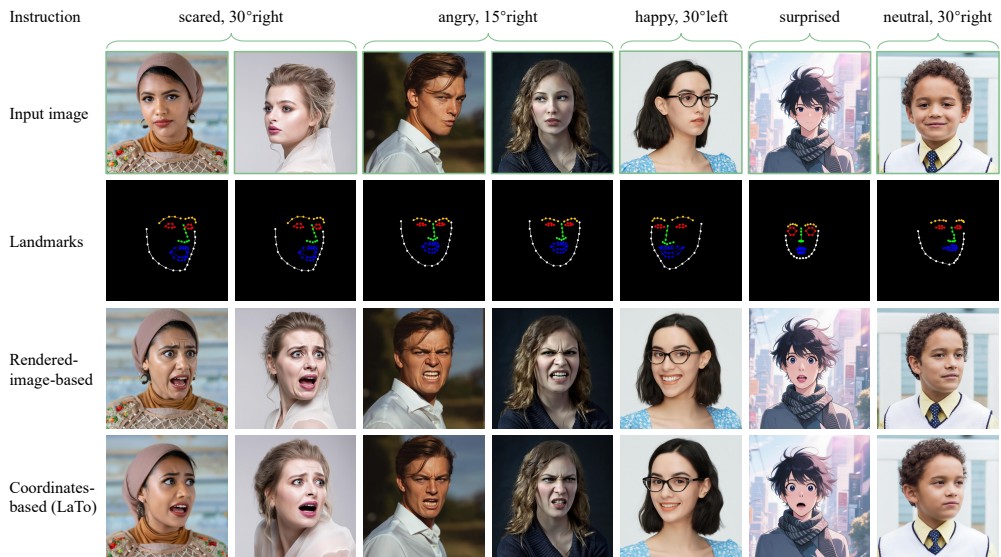

Figure 1: Landmark tokenization in LaTo preserves identity and produces natural results, whereas pixelwise alignment baselines rigidly follow the rendered landmark image and often lose identity under cross-identity landmark conditions (first four columns) or when self-identity landmarks differ substantially from the source.

copying instead of geometric reasoning, causing identity drift when landmarks differ greatly from the source, and also increases memory and computation because dense token sequences interact through quadratic self-attention (Huang et al., 2024; Avrahami et al., 2025).

In this paper, we present LaTo, a landmark-tokenized Diffusion Transformer for complex face editing. Instead of using dense pixel-wise landmark renderings, LaTo quantizes landmark coordinates into discrete facial tokens via a VQ-VAE-style codebook (Van Den Oord et al., 2017), producing embeddings aligned with image-token dimensionality while preserving facial structure. To route these spatially discontinuous tokens to the correct facial regions, we propose location-mapping positional encoding, which anchors each token to its physical location in the latent grid for precise regional guidance. Following Step1X-Edit, we feed landmark tokens as context into DiT blocks for unified token processing, enabling flexible yet decoupled interactions among geometry, appearance, and instruction with high efficiency, strong identity preservation, and semantic consistency. We further introduce landmark-aware classifier-free guidance to balance visual quality and geometric fidelity.

To address training data limitations, we develop an automated synthesis-and-curation pipeline to construct HFL-150K, a large-scale dataset of more than 150,000 face editing pairs with diverse attributes and strict identity consistency. Each pair is annotated with fine-grained editing instructions and high-precision facial landmarks, providing the rich supervision necessary to fully realize LaTo's capabilities. At inference, supplying precise landmark inputs can be impractical for end users. To alleviate this requirement, we introduce a landmark predictor that employs a vision–language model (VLM) to infer target landmarks from the source image and textual instruction using a structured chain-of-thought. We collect a set of high-quality instruction–landmark annotations with explicit change magnitudes and fine-tune a lightweight VLM, substantially improving landmark estimation accuracy and usability. Equipped with HFL-150K and the landmark predictor, LaTo achieves state-of-the-art face editing performance, particularly in semantic consistency and identity preservation. Our contributions can be summarized as follows:

- We introduce HFL-150K, a face-editing dataset comprising over 150K face pairs annotated with fine-grained editing instructions. To the best of our knowledge, it is the largest resource in this area.

- We present LaTo, the first landmark-tokenized diffusion transformer for precise face editing that integrates (i) landmark tokenization of raw coordinates, (ii) location-mapping positional encoding, and (iii) landmark-aware classifier-free guidance. This design affords

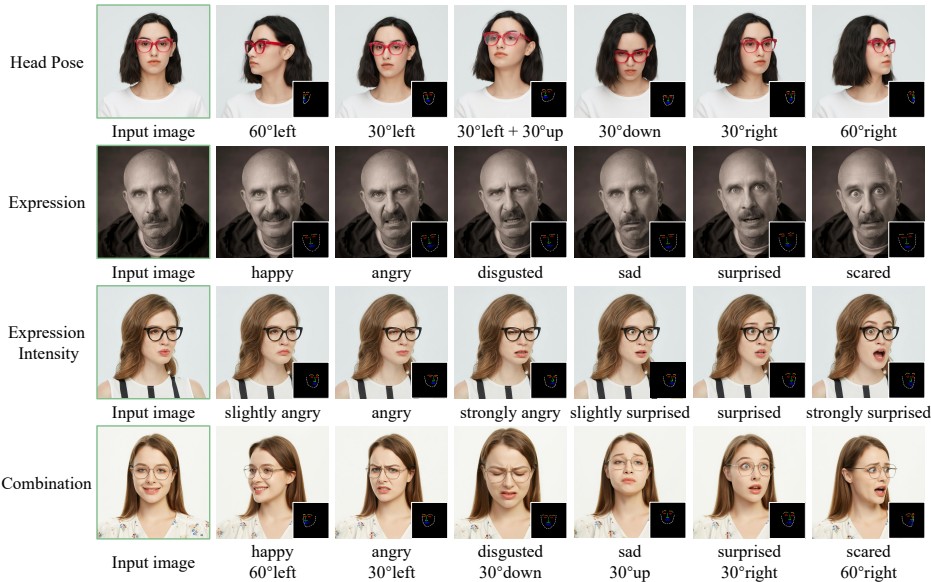

Figure 2: LaTo enables fine-grained facial expression editing, parametric head-pose editing, or their combination. The small images visualize generated landmarks via landmark predictor, enabling intuitive control signal acquisition.

flexible geometric control, strong identity preservation, and reduced computational cost compared with rendered-image conditioning.

- We develop a landmark predictor—a lightweight VLM that infers target landmarks from a source image and an editing instruction, bridging semantics and precise facial geometry for intuitive user control.

- As shown in Figure 2, leveraging HFL-150K and the Landmark Predictor, LaTo delivers precise facial attribute control under complex, fine-grained instructions and achieves state-of-the-art performance.

## 2 RELATED WORK

### 2.1 INSTRUCTION-BASED IMAGE EDITING MODELS

Diffusion models are the de facto standard for high-fidelity text-to-image synthesis and the foundation of many instruction-driven editing systems. Existing methods are either training-free, manipulating the reverse process through latent inversion (Tumanyan et al., 2023; Rombach et al., 2022; Mokady et al., 2023) or attention control (Cao et al., 2023; Wang et al., 2024), or training-based, fine-tuning on large-scale paired data for stronger performance. InstructPix2Pix (Brooks et al., 2023) introduced synthetic supervision, while later methods improve instruction grounding by integrating VLMs with diffusion models, e.g., MGIE (Fu et al., 2023), Emu Edit (Sheynin et al., 2024), SmartEdit (Huang et al., 2024), AnyEdit (Yu et al., 2025), UltraEdit (Zhao et al., 2024), OmniGen (Xiao et al., 2025), BAGEL (Deng et al., 2025), and ACE (Han et al., 2024), or by injecting VLM latents into diffusion decoders (Chen et al., 2025; Pan et al., 2025a; Liu et al., 2025). Generalist systems such as GPT-4o (Hurst et al., 2024) and Gemini (Comanici et al., 2025) also show strong vision–language coherence. However, precise spatial alignment and identity preservation in face editing remain challenging because most methods rely on semantic cues rather than explicit geometry.

### 2.2 FACE EDITING MODELS

Face editing aims to alter facial attributes while preserving identity (Preechakul et al., 2022). Recent text-driven methods, such as StyleCLIP (Patashnik et al., 2021) and ChatFace (Yue et al., 2023),

Table 1: Key attributes of human face editing benchmarks. HFL-150K surpasses existing face benchmarks in both scale and diversity, with unique strengths in fine-grained instruction alignment.

| Benchmarks | Size | Real Image | Training | Fine-grained Instruction | Expression | Head pose |
|---|---|---|---|---|---|---|
| ICE-Bench | 206 | ✓ | ✗ | ✗ | ✓ | ✗ |
| SeqDeepFake | 49,920 | ✗ | ✓ | ✗ | ✓ | ✗ |
| SEED | 91,526 | ✗ | ✓ | ✗ | ✓ | ✓ |
| **HFL-150K** | 302,014 | ✓ | ✓ | ✓ | ✓ | ✓ |

achieve strong visual quality but often yield entangled edits and unintended identity shifts, especially under large instruction changes. To improve control and anatomical consistency, later works introduce structured conditions such as face masks (Zhang et al., 2025), semantic layouts (Mofayezi et al., 2024), and landmark images (Li et al., 2022; Sun et al., 2024). In DiT-based editing systems (Tan et al., 2024; Pan et al., 2025a), landmarks are usually rasterized and encoded by a visual VAE to guide diffusion. While effective for geometric alignment, this pixel-wise conditioning can encourage template copying, induce identity drift when geometry changes substantially, and increase token length and computational cost. LaTo addresses these issues by directly modeling landmark coordinates and their corresponding facial regions, decoupling geometry from pixel-level appearance.

## 3 METHODOLOGY

### 3.1 HFL-150K DATASET CONSTRUCTION

Large-scale datasets are essential for face editing. Existing benchmarks such as SeqDeepFake (Shao et al., 2022) and SEED (Zhu et al., 2025) are limited by coarse-grained instructions, outdated synthesis models (Karras et al., 2019; Tsaban & Passos, 2023), and substantial low-quality samples, leading to artifacts, limited diversity, and restricted scale. To overcome these issues, we introduce HFL-150K, a 150k-scale face editing dataset of source image, instruction, and edited image triplets, constructed via a hybrid pipeline of real-image curation and synthetic generation with advanced editing models.

**Fine-grained attribute definition.** We focus on two core facial editing tasks: expression editing (e.g., "make him smile gently") and parametric head pose editing (e.g., "rotate her head 30° left"). For expression categorization, we adopt standard emotion recognition protocols (Huang et al., 2023) to define seven canonical expressions and assign intensity levels (slightly, normally, strongly) based on visual saliency. For head pose parameters, we formulate spatial transformations using yaw and pitch angles with 30° as the base unit of motion amplitude, as shown in Figure 3 (c–d).

**Synthetic data collection.** As shown in Figure 3 (a), we employ advanced editing models (Step1X-Edit, GPT-4o (Hurst et al., 2024), BAGEL (Deng et al., 2025), and FLUX.1-Kontext) to generate a synthetic dataset focusing on either expression or head pose edits, aligning with their single-turn training objectives. To ensure generation quality, we implement instruction-specific filtering: (1) For expression edits, an expression validator computes semantic similarity between generated outputs and input instructions using Qwen2.5-VL (Bai et al., 2025). (2) For pose edits, a pose discriminator estimates head orientation via Euler angle regression (Yang et al., 2019) from facial landmarks and verifies alignment with target rotations. Only validated samples are retained, resulting in a **34K-sample** dataset. This synthetic dataset provides a simplified prior that enhances model interpretability.

**Real-world data collection.** As illustrated in Figure 3 (b), we further construct a real-world face subset from human-centric video datasets (Li et al., 2025), leveraging natural dynamics to capture intra-identity variation in expression and head pose. We apply a multi-stage filtering process comprising: (1) a quality filter using a face detector (Deng et al., 2019) to drop occlusions and enforce centering, a Laplacian of Gaussian (LoG) to remove motion blur, and Dover (Wu et al., 2023) for aesthetic and technical assessment; (2) a diversity filter combining geometric analysis (2D facial landmarks (Zhu et al., 2016)) and semantic analysis (Qwen2.5-VL for high level facial changes). Pairs with scores beyond thresholds are removed, including abnormally high values indicating copy-

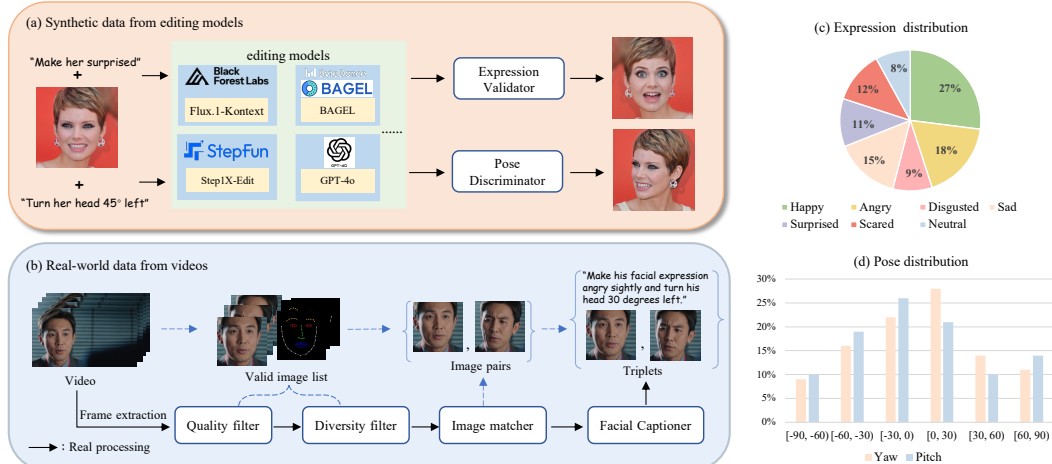

Figure 3: Data collection pipeline and statistics of HFL-150K. (a) Synthetic data generation via advanced editing models. (b) Real image pair extraction from video datasets. (c) Expression distribution across 7 categories. (d) Head pose angles aligned with 30° motion budgets.

paste artifacts and critically low values caused by variability between different people. Finally, an image matcher with CLIP (Radford et al., 2021) performs cross frame identity verification, removing residual pairs from different people. Through this pipeline, a total of **116K** high-quality, semantically diverse image pairs are generated, reflecting natural facial variations.

For deriving fine-grained editing instructions, we leverage a facial captioner to recognize expressions and estimate intensity. However, even advanced Qwen2.5-VL-72B shows limitations in fine-grained expression recognition. To address this, we curate a high-quality dataset and fine-tune the model to improve sensitivity to more subtle magnitudes. We manually annotate 18k samples with seven predefined expression categories and intensity levels (see Appendix for guidelines). For head pose estimation, we use both horizontal and vertical optical flow angles (Karaev et al., 2024) to quantify motion amplitude relative to our base unit. A unified template is designed for instruction generation:

*Make his/her facial expression {expression-type} {intensity-level} and turn his/her head {angle-degree}.*

## 3.2 LaTo

Building upon the Step1X-Edit, we propose LaTo, a fine-grained human face editing framework that effectively leverages compact facial tokens. As illustrated in Figure 4, LaTo achieves precise and user-friendly face editing through three core mechanisms: landmark tokenizer, multi-modal token fuser and landmark predictor.

### 3.2.1 Landmark Tokenizer

Building on the widely adopted VQVAE architecture for discrete tokenization, the landmark tokenizer combines an encoder-decoder framework with a lightweight quantizer. Given a raw landmark sequence $F = \{(X_i, Y_i)\}_{i=1}^n$ (with $n$ 2D locations), the encoder maps it into a continuous latent space $E \in R^{n \times d}$ via residual blocks with convolutions. A quantizer then discretizes these latents through nearest-neighbor lookup in a learnable codebook $C \in R^{m \times d}$ of size $m$, generating compact yet expressive facial tokens in a unified geometric space. The decoder, structured to mirror the encoder, reconstructs the input sequence, ensuring spatial coherence. The complete training objective combines reconstruction loss and commitment loss:

$$\mathcal{L} = \|F - \hat{F}\|_1 + \beta \|E - \text{sg}[C]\|_2^2 \tag{1}$$

where sg[·] denotes the stop-gradient operation, and $\beta$ is the weight of the commitment loss. This loss encourages faithful reconstruction while promoting effective codebook utilization, enabling the model to learn both geometric accuracy and semantic expressiveness in facial token representations.

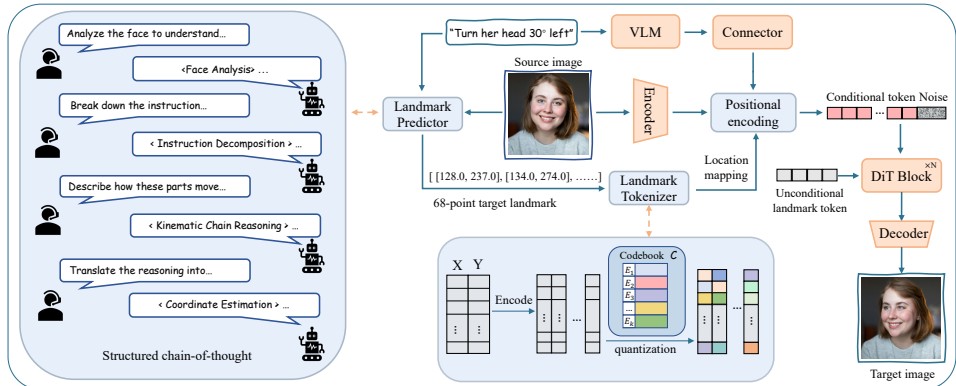

Figure 4: Overview of LaTo. LaTo predicts target landmarks from the source image and instruction, then tokenizes the landmarks and source image with a landmark tokenizer and visual VAE. Location-mapping positional encoding aligns landmark tokens with their physical positions, while learned unconditional landmark tokens guide denoising to keep edits consistent with both landmarks and instruction.

### 3.2.2 MULTI-MODAL TOKEN FUSER

We develop a token fuser to flexibly integrate landmark, image, and semantic tokens through three components: location-mapped landmark positional encoding, unified representation and landmark-aware classifier-free guidance.

**Location-mapping landmark positional encoding.** Step1X-Edit employs 3D Rotary Positional Encoding (RoPE) (Su et al., 2024) to encode spatial information for both image and text tokens. For each position $i$ in image tokens, the 3D RoPE is computed as:

$$P_i = \text{Concat}\left(R_T(0),\ R_H\left(\left\lfloor \frac{i}{h} \right\rfloor\right),\ R_W(i\%h)\right), \tag{2}$$

where $h$ represents the height of the latent grid. Here, each $R(\cdot)$ implements 1D rotary embeddings applied to the text, height, and width dimensions. These embeddings are independently repeated across spatial axes to model positional relationships between tokens. To maintain spatial consistency, we design a location mapping mechanism that links landmark tokens to their physical location in the latent grid:

$$P_i = \text{Concat}\left(R_T(0),\ R_H\left(y_i\right),\ R_W(x_i)\right), \tag{3}$$

where $(x_i, y_i)$ denote downsampled coordinates of landmark points from the original image. This ensures each compressed representation accurately guides its corresponding area, preserving conditioning fidelity. Our experiments 4.3 demonstrate that, in the absence of this correction, the model struggles to learn spatial relationships between landmark inputs and generated tokens, leading to blurry or misaligned facial features.

**Unified representation.** Following Step1X-Edit, we treat all modal tokens as a unified representation. A trainable facial landmark adapter first projects facial tokens into the same latent space as noisy tokens, denoted as $z_f \in \mathbb{R}^{l_f \times d}$. The input is formulated as:

$$\mathbf{Z} = \text{Concat}(z_\text{t}, z_\text{s}, z_f, z_\text{n}) \in \mathbb{R}^{(l_t + l_s + l_f + l_n) \times d}, \tag{4}$$

where $z_\text{t} \in \mathbb{R}^{l_t \times d}$, $z_\text{s} \in \mathbb{R}^{l_s \times d}$, and $z_\text{n} \in \mathbb{R}^{l_n \times d}$ denote semantic text tokens, source visual tokens, and noisy image tokens, respectively. The multi-modal attention mechanism generates query and key via:

$$\begin{aligned} \mathbf{P} &= \text{Concat}(P_\text{t}, P_\text{s}, P_f, P_\text{n}) \\ \mathbf{Q} &= \text{RoPE}(W_q(\mathbf{Z}), \mathbf{P}), \quad \mathbf{K} = \text{RoPE}(W_k(\mathbf{Z}), \mathbf{P}), \end{aligned} \tag{5}$$

with $P_\text{t}, P_\text{s}, P_\text{n}$ derived from Equation 2 and $P_f$ from Equation 3. This formulation enables flexible token interactions via DiT's multi-modal attention mechanism, allowing direct relationships between

Table 2: Quantitative evaluation of state-of-the-art editing methods on HFL-150K and face attribute editing subsets from GEdit-Bench/ICE-Bench. † Indicates models fine-tuned on HFL-150K.

| Method | HFL-150K | | | | GEdit&ICE-Bench(Subset) | | | |
|---|---|---|---|---|---|---|---|---|
| | SC ↑ | VQ ↑ | NA ↑ | IP ↑ | SC ↑ | VQ ↑ | NA ↑ | IP ↑ |
| Instruct-PixPix (Brooks et al., 2023) | 0.518 | 0.582 | 0.675 | 0.381 | 0.573 | 0.567 | 0.643 | 0.405 |
| AnyEdit (Yu et al., 2025) | 0.612 | 0.641 | 0.702 | 0.446 | 0.669 | 0.654 | 0.676 | 0.479 |
| OmniGen (Xiao et al., 2025) | 0.737 | 0.688 | 0.731 | 0.503 | 0.755 | 0.707 | 0.697 | 0.536 |
| Bagel (Deng et al., 2025) | 0.786 | 0.709 | 0.759 | 0.539 | 0.797 | 0.718 | 0.733 | 0.579 |
| Step1X-Edit (Liu et al., 2025) | 0.751 | 0.706 | 0.732 | 0.518 | 0.767 | 0.694 | 0.705 | 0.541 |
| Step1X-Edit† (Liu et al., 2025) | 0.804 | 0.725 | 0.801 | 0.571 | 0.803 | **0.732** | 0.788 | 0.594 |
| FLUX.1-Kontext (Labs et al., 2025) | 0.712 | 0.720 | 0.779 | 0.556 | 0.749 | 0.693 | 0.751 | 0.576 |
| FLUX.1-Kontext† (Labs et al., 2025) | 0.786 | 0.737 | **0.816** | 0.593 | 0.801 | 0.713 | 0.771 | 0.609 |
| **LaTo (ours)** | **0.832** | **0.749** | 0.805 | **0.634** | **0.829** | 0.724 | **0.793** | **0.651** |

any token pair without rigid spatial constraints. Given the facial landmark token length $l_f = 68$ is significantly smaller than noisy image tokens ($l_n = 1024$), this approach maintains computational efficiency comparable to the baseline model.

**Landmark-aware Classifier-free guidance.** To balance image quality and landmark fidelity, we propose landmark-aware classifier-free guidance (CFG). In standard image-conditioned pipelines, the unconditional branch uses a zero image. Similarly zeroing landmark coordinates, however, especially at high CFG weights, often causes reference-face copying and suppresses appearance changes correlated with landmarks. We argue that zeroed landmark embeddings do not represent an unconstrained geometric state, but instead conflict with facial motion dynamics, creating undesired coupling even in the unconditional branch. Inspired by MTVCrafter (Ding et al., 2025), we replace position-aware landmark tokens with learnable unconditional tokens during unconditional passes, yielding a more meaningful unconditional distribution and improved robustness.

## 3.3 LANDMARK PREDICTOR

To enable intuitive interaction and accurate landmark estimation, we fine-tune Qwen2.5-VL-3B as a landmark predictor (LP). Given a source image and instruction, LP predicts target landmark coordinates via structured CoT reasoning Qiang et al. (2026); Gu et al. (2026), including: (1) initial analysis of pose, expression, and landmark alignment; (2) decomposition of the instruction into anatomical motions; (3) kinematic reasoning over rigid motions and non-rigid deformations; and (4) coordinate estimation on a normalized 512×512 canvas, producing a machine-parsable list of $(X, Y)$ pairs. We build CoT supervision for 23,145 HFL-150K triplets with a rule-guided pipeline and retain 19,398 manually verified examples for fine-tuning. The model fuses visual and textual inputs in a multimodal transformer and is trained with next-token prediction to generate structured CoT sequences. A compact coordinate tokenization scheme and fixed output grammar improve numerical fidelity, while inference-time smoothing and geometric sanity checks preserve identity-consistent rigid distances. This design provides interpretable and precise landmark predictions for instruction-guided geometric control in face editing.

## 4 EXPERIMENTS

### 4.1 EXPERIMENTAL SETUP

**Implement details.** The landmark tokenizer is trained from scratch on HFL-150K with an 8,192-entry codebook and 3,072-dimensional codes matched to the Step1X-Edit hidden size. Training uses 8 NVIDIA A100 GPUs with batch size 128 per GPU for 100k iterations, and unused codes are reset every 50 steps to prevent saturation. For LaTo, we fine-tune the base model with LoRA (rank 64) and add an unfrozen linear landmark adapter. The model is trained on 16 NVIDIA A100 GPUs with total batch size 32, learning rate $1 \times 10^{-4}$. We replace conditional landmark tokens with unconditional ones with probability 0.1 to encourage diversity, and train for a total of 40k iterations.

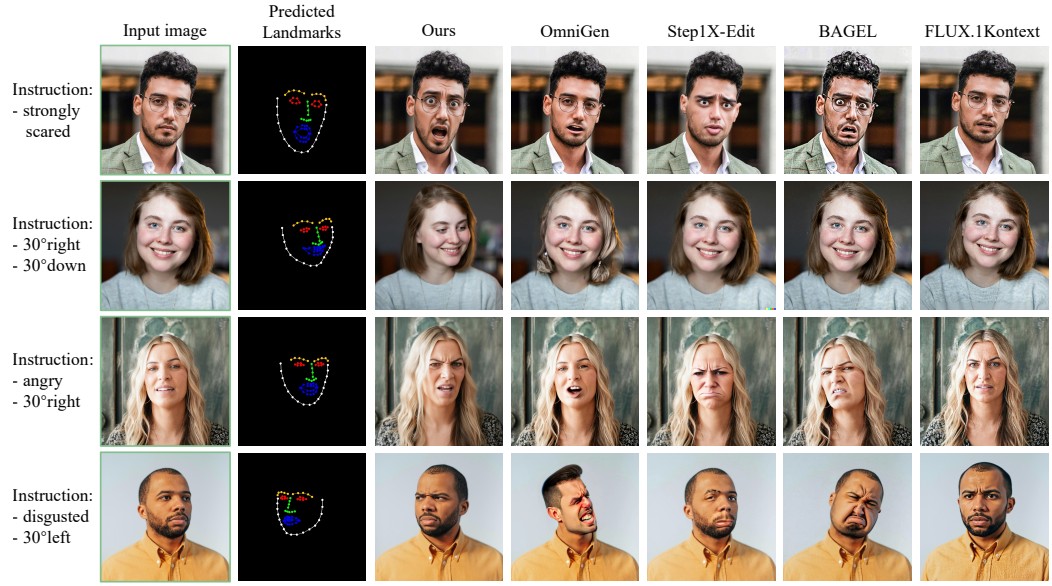

Figure 5: Qualitative comparison with state-of-the-art image editing methods.

**Dataset.** We split HFL-150K into 150,000 training samples and 1,007 test samples. For the test set, stratified sampling ensures diverse coverage of expression types, head poses, and their combinations. We additionally perform manual validation to ensure all samples include high-quality instructions and image–landmark pairs. We also curate an auxiliary test set by selecting face-attribute editing samples from GEdit-Bench (Liu et al., 2025) and ICE-Bench (Pan et al., 2025b), yielding 103 samples for further evaluation. Detailed curation protocols are provided in the Appendix.

**Metrics.** We propose a four-criteria framework to evaluate both identity preservation and accuracy of requested modifications, comprising Semantic Consistency (SC), Visual Quality (VQ), Natural Appearance (NA), and Identity Preservation (IP). SC, VQ, and NA are scored by Qwen2.5-VL-72B using a normalized $[0, 1]$ scale via visual reasoning. For IP, we first use ArcFace similarity $s_{arc}$ (Deng et al., 2019), but some methods inflate it by copying or barely altering the source. We therefore define a rectified score that penalizes such cases: Qwen2.5-VL parses the source face and predicts expected edit amplitude $\varphi_{ins}$ from the instruction, while the actual amplitude $\varphi_{real}$ is derived from SSIM between source and edit. The rectified IP score calculated as:

$$s_{rip} = \max(0, s_{arc} - (\frac{\varphi_{ins} - \varphi_{real}}{\varphi_{ins} + \epsilon})^2), \qquad (6)$$

## 4.2 QUALITATIVE AND QUANTITATIVE ANALYSIS

We benchmark LaTo on HFL-150K and the face-attribute splits of GEdit-Bench/ICE-Bench. For fairness, we fine-tune Step1X-Edit and Kontext on HFL-150K using their official implementations. Table 2 shows significant gains across all metrics. Fine-tuning on HFL-150K improves baseline SC by 5.3% and 7.4%, suggesting that the dataset better captures real-world diversity. On HFL-150K, LaTo surpasses Bagel by 4.6% SC, while LaTo-IP outperforms FLUX.1-Kontext by 7.8%. On GEdit-Bench/ICE-Bench, LaTo exceeds Bagel by 7.2%, indicating stronger identity preservation. Despite using the same training data and base model as Step1X-Edit†, LaTo still gains 2.9% on average across all metrics. Figure 5 further shows better pose control, identity consistency, and photorealism under large expression changes.

## 4.3 ABLATION STUDY

**Facial condition formulations.** We compare LaTo with two image-based landmark conditioning schemes: rendered landmark images and a compressed variant with position shifting (Tan et al.,

Table 3: Ablation study on landmark condition types and landmark positional encoding. Landmark alignment is measured by Landmark Error (LE), the L1 distance between edited-image landmarks and provided landmarks.

| Landmark type | Encoder | Additional Setting | SC ↑ | VQ ↑ | NA ↑ | IP ↑ | Latency(s) ↓ | Landmark Error ↓ |
|---|---|---|---|---|---|---|---|---|
| / | / | / | 0.804 | 0.725 | 0.801 | 0.571 | **49.6** | - |
| rendered image | visual VAE | / | 0.821 | 0.712 | 0.744 | 0.584 | 83.6 | **1.76** |
| | | compression | 0.816 | 0.704 | 0.709 | 0.569 | 61.3 | 3.07 |
| coordinates | landmark tokenizer | (1) w/o PE | 0.654 | 0.611 | 0.630 | 0.512 | 50.7 | 58.9 |
| | | (2) RoPE | 0.778 | **0.751** | 0.786 | 0.621 | 52.1 | 25.4 |
| | | (3) learnable RoPE | 0.803 | 0.737 | 0.791 | 0.617 | 52.9 | 9.63 |
| | | (4) ours | **0.832** | 0.749 | **0.805** | **0.634** | 52.1 | 2.34 |

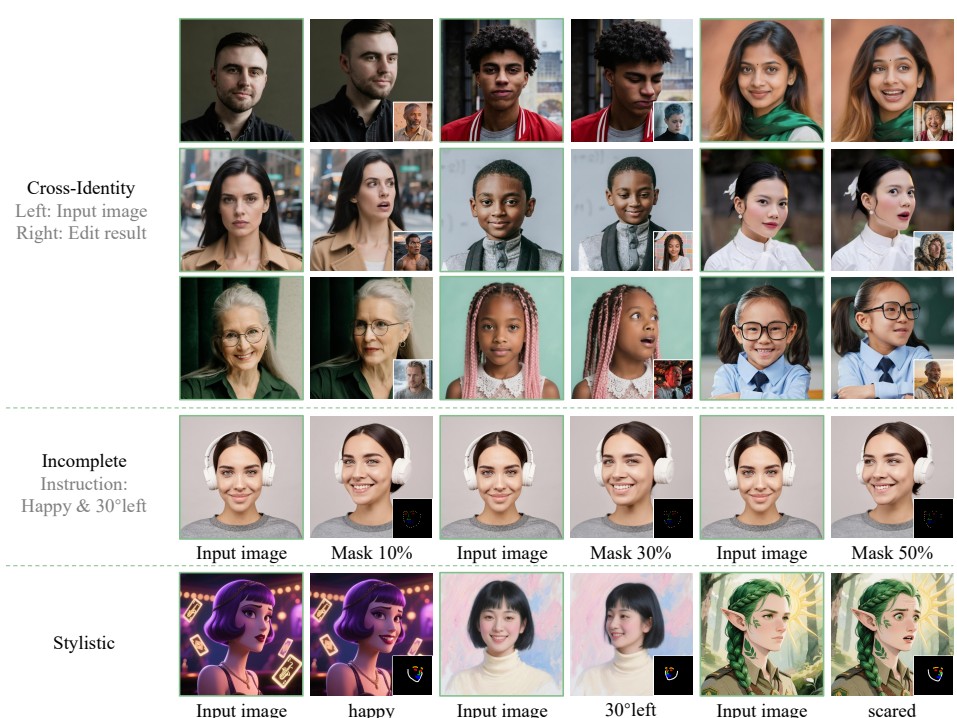

Figure 6: Qualitative results on challenging inputs. For cross-identity cases, the corresponding driving images are shown in the bottom-right corner.

2025). Table 3 shows that both image-based designs remain limited: relative to the fine-tuned baseline, NA drops by 5.7%, IP improves by only 1.3%, and compression further degrades performance despite a 26% speedup. In contrast, our landmark tokenization explicitly models coordinate-to-attribute relations and decouples control strength, yielding 6.1% higher NA, 1.1% higher SC, and 5.0% higher IP than rendered-image conditioning. It also achieves a 37% speedup, matching the baseline in efficiency.

**Landmark positional encoding effectiveness.** We compare positional encodings for landmark tokens, including original RoPE, learnable RoPE, no PE, and our location-mapping RoPE. Table 3 shows that no PE causes unstable training and unnatural results, while original RoPE poorly captures landmark–image spatial relations, yielding a landmark error of 25.4. Learnable RoPE improves instruction following and landmark conditioning, but still underperforms our method. In contrast, location-mapping RoPE provides a stronger geometric prior, enabling better geometry extraction and the best landmark conditioning and identity fidelity.

**Landmark predictor accuracy evaluation.** To evaluate the effectiveness of landmark predictor, we conducted a manual accuracy assessment against Gemini 2.5 Pro, Qwen2.5-VL-72B, and Qwen2.5-

Table 4: Comparison of landmark predictor performance using unified prompting conditions.

| Methods | Accuracy |
|---|---|
| Qwen2.5-VL-3B | 0.477 |
| Qwen2.5-VL-72B | 0.597 |
| Gemini 2.5 Pro | 0.613 |
| **Ours** | **0.730** |

Table 5: Performance analysis across different CFG scales for landmark condition.

| CFG-scale | SC ↑ | VQ ↑ | NA ↑ | IP ↑ | LE ↓ |
|---|---|---|---|---|---|
| 1 | 0.803 | 0.733 | **0.816** | 0.619 | 9.63 |
| 4 | **0.832** | **0.749** | 0.805 | **0.634** | 2.34 |
| 7 | 0.821 | 0.698 | 0.751 | 0.616 | 1.94 |
| 10 | 0.807 | 0.656 | 0.679 | 0.588 | **1.82** |

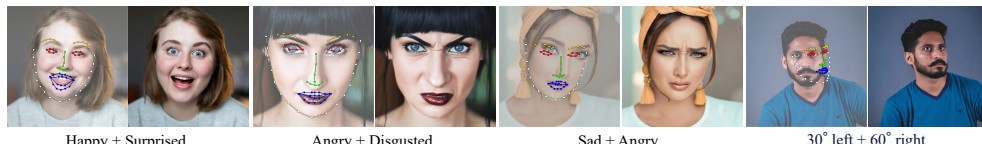

Happy + Surprised     Angry + Disgusted     Sad + Angry     30° left + 60° right

Figure 7: Visualization of the performance of landmark predictor under ambiguous instructions. Predicted landmarks are overlaid on the source images.

VL-3B. Specifically, we selected 50 human faces from the HFL-150K test set and randomly generated instructions for each image by altering expression, head pose angle, or their combinations (6 variations per image), resulting in 300 samples. The predicted landmarks were rendered on source images, and participants were asked to evaluate whether the predicted landmarks aligned with the given instructions, assigning a score of 0 (incorrect) or 1 (correct). We recruited 10 human evaluators and collected their results, which are presented in Table 4. Our fine-tuned model achieves the highest average accuracy among the compared models, outperforming the baseline by 25.3%, demonstrating its superiority in landmark analysis. To further evaluate LP under realistic noisy instructions, we test it on composite expressions (e.g., happy + surprised) and conflicting pose instructions (e.g., turn left and turn right), as shown in Figure 7.

**Landmark-aware CFG scaling analysis.** Table 5 illustrate the impact of our CFG scale. Raising the CFG scale improves landmark alignment, yet may generate more artifacts and compromise video quality. We adopt a CFG scale of 4 as the optimal baseline configuration.

**The sensitivity to challenging inputs.** Although LaTo is primarily self-landmark-driven via the proposed LP, we further evaluate it under three challenging settings: missing landmarks, cross-identity landmarks, and stylized inputs. As shown in Figure 6, LaTo largely preserves identity under cross-identity conditioning and moderate landmark dropout, while maintaining stable editing quality on non-photorealistic inputs. We attribute this robustness to three components: the landmark tokenizer decouples shape from appearance for more stable geometric reasoning; the token fuser jointly models geometry, instructions, and identity for implicit feature-level control; and learned unconditional landmark tokens adaptively balance geometry and identity, allowing LaTo to retain basic visual quality and key facial traits even with severe landmark loss.

## 5 CONCLUSION

We presented LaTo, a landmark-tokenized diffusion transformer for fine-grained, identity-preserving face editing. LaTo quantizes landmark coordinates into discrete facial tokens and aligns them with image tokens via a location mapping positional encoding, which decouples geometry from appearance and enables precise control with strong identity preservation and high efficiency. A vision–language landmark predictor with structured reasoning infers target landmarks from instructions and source images, improving robustness and interactive controllability. This design removes the need for dense pixel-wise correspondence and mitigates identity drift under large pose or expression changes. To support research at scale, we curate HFL-150K, a large-scale benchmark of face pairs with fine-grained instructions, spanning real-world imagery and outputs from advanced models. Extensive experiments demonstrate that LaTo delivers state-of-the-art photorealism, semantic consistency, and computational efficiency, establishing a strong foundation for controllable, human-centric editing.

ACKNOWLEDGEMENTS

This work is supported by National Natural Science Foundation of China (62571322, 62431015, 62271308), STCSM (24ZR1432000, 24511106902, 24511106900, 22DZ2229005), 111 plan (BP0719010), and State Key Laboratory of UHD Video and Audio Production and Presentation.

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
