# OpenReview forum: "LaTo:  Landmark-tokenized Diffusion Transformer for Fine-grained Human Face Editing"
_ICLR.cc/2026/Conference — ICLR 2026 Poster_

### Official Review · Reviewer_bTdN · 2025-10-23

**Soundness:** 3
**Presentation:** 3
**Contribution:** 3
**Rating:** 6
**Confidence:** 5

**Summary:**

This paper introduces LaTo, a novel face-editing framework that leverages landmark-tokenized diffusion transformers (DiTs) for identity-preserving and fine-grained facial attribute control. The model quantizes facial landmark coordinates into discrete tokens and uses a landmark tokenizer along with a location-mapping positional encoding. LaTo aims to overcome challenges faced by existing methods by improving computational efficiency, identity preservation, and semantic consistency. The paper further introduces a new dataset, HFL-150K, containing over 150,000 face pairs with fine-grained editing instructions, which is used to train and benchmark LaTo.

**Strengths:**

- Landmark tokenization is an innovative approach that moves beyond traditional pixel-based conditioning in face editing tasks. This method helps prevent identity drift when landmarks differ significantly from the source image.
- The HFL-150K dataset is a valuable contribution, offering a large collection of high-quality real-world face pairs with fine-grained instructions essential for training advanced face editing models.
- Comprehensive ablation studies and both qualitative and quantitative evaluations demonstrate LaTo’s effectiveness across multiple metrics, supporting its proposed innovations.

**Weaknesses:**

- The paper mainly focuses on facial expression and head pose editing but provides limited discussion on how well LaTo generalizes to other image domains or tasks (e.g., full-body or complex scene editing). Assessing its broader applicability would strengthen the claims.
- Although HFL-150K is extensive, there is little information about its diversity regarding age, ethnicity, or other demographics—factors crucial for ensuring robust performance across different populations. More details on how these aspects were considered during dataset construction would enhance transparency.
- There is an inconsistency: cfg is mentioned in the Methods section but not in the Abstract.
- The main contribution is not included in the framework, making it confusing and difficult to understand.

**Questions:**

- Can the model be optimized for real-time use in applications like live avatar creation or video conferencing? What specific optimizations are needed to enable efficient on-the-fly processing?
- How well does LaTo perform across various demographic groups such as age, ethnicity, or gender? Could you provide an in-depth analysis of its results on more diverse faces to ensure fair representation?
- While the paper notes limitations of pixel-wise methods when landmarks deviate from the source image, how does LaTo handle extreme cases where landmark positions are highly inaccurate due to large pose changes, occlusions, or exaggerated expressions? Is there any noticeable degradation in output quality under these conditions?

---

> ### Author Response · Authors · 2025-11-21
> **Response to Reviewer bTdN (Part 1/2)**
>
> **[Q1]: Can the model be optimized for real-time use in applications like live avatar creation or video conferencing? What specific optimizations are needed to enable efficient on-the-fly processing?**
>
> **[A1]:** For video applications, in our follow-up work we extend LaTo to facial animation based on Wan2.2, enabling video-avatar generation. As shown in Figure 11, LaTo still preserves identity robustly when editing with cross-identity reference face videos, which demonstrates strong scalability.
>
> Regarding on-the-fly processing, we incorporate several acceleration strategies into LaTo, and the trade-offs between speed and performance on HFL-150K are summarized in the table below. Under the most aggressive configuration, the processing time is reduced to 4.9 seconds per image on two A100 GPUs while largely preserving editing quality, although this is still significantly below real-time performance.  The bottleneck is the base model Step1X-Edit, as the additional cost from our token fuser and landmark predictor is minimal, requiring only 1.83 s and 0.67 s respectively under the default configuration. Bridging the remaining gap will likely depend on future advances in accelerating such base models by the broader research community.
>
> | Methods                           | SC      | VQ      | NA      | IP      | Latency(s) |
> |-|-|-|-|-|-|
> | Default (1024px)                  | **0.832** | **0.749** | **0.805** | **0.634** | 52.1    |
> | 512px                             | 0.820   | 0.725   | 0.790   | 0.619   | 11.9      |
> | 512px + TeaCache                  | 0.814   | 0.715   | 0.771   | 0.607   | 8.5       |
> | 512px + TeaCache + xDiT(GPU=2)    | 0.814   | 0.713   | 0.774   | 0.608   | **4.9**       |
>
>
> **[Q2]: How well does LaTo perform across various demographic groups such as age, ethnicity, or gender? Could you provide an in-depth analysis of its results on more diverse faces to ensure fair representation?**
>
> **[A2]:** We appreciate this important observation! Following standard practice [1], we estimated age, gender, and skin tone using Qwen3‑VL. The results show that HFL‑150K contains 34.2% subjects aged 40 or above, 48.3% female and 51.7% male, and 23.1% dark‑skinned individuals, with the rest distributed across medium (45.7%) and light (31.2%) skin tones. The qualitative results in Figure 6 show that LaTo generalizes well across demographic groups, supported by the diversity of HFL‑150K. Furthermore, LaTo maintains stable editing performance even for stylistically different inputs such as cartoons and oil paintings.

---

> > ### Author Response · Authors · 2025-11-21
> > **Response to Reviewer bTdN (Part 2/2)**
> >
> > **[Q3]: While the paper notes limitations of pixel-wise methods when landmarks deviate from the source image, how does LaTo handle extreme cases where landmark positions are highly inaccurate due to large pose changes, occlusions, or exaggerated expressions? Is there any noticeable degradation in output quality under these conditions?**
> >
> > **[A3]:** We treat landmark conditions as extreme in two cases: (1) cross-identity landmarks with large discrepancies in pose or expression, and (2) missing landmarks caused by occlusion. To assess robustness, we collected 20 portraits spanning diverse ages, genders, and ethnicities. For each portrait, we sampled 10 cross-identity landmarks and generated self-driven landmarks with LP, then randomly dropped a subset, which we replaced with unconditional landmark tokens to simulate missing points. As shown in the table below, both cross-identity conditioning and moderate landmark dropout cause only minor degradation in identity preservation, and LaTo still outperforms the baseline by about 3% in overall rating.  We attribute this robustness mainly to three factors. First, the landmark tokenizer decouples explicit, shape-related structure from pixel-level appearance, which stabilizes geometric reasoning and helps the model capture more generalizable expression dynamics and identity-agnostic pose variations in cross-identity settings. Additional experiments discussed in Reviewer oX4b’s Q5 further demonstrate that quantization leads to more unified facial representations and improved generalization. Second, the token fuser encourages the model to jointly interpret driving facial geometry, editing instructions, and visual identity features. This coordinated interpretation balances their respective contributions and enables geometry-driven control in an implicit, feature-level manner, rather than relying on explicit geometric tracking.  Third, the learned unconditional landmark tokens provide each landmark position with a flexible way to adjust the coupling between geometry and identity. Even when landmark absence becomes severe (50%), LaTo, while showing noticeable degradation in identity preservation, still maintains basic visual quality and key facial traits (Figure 6).
> >
> > | Methods                         | SC                | VQ                | NA                | IP                |
> > |-|-|-|-|-|
> > | Step1X-Edit†                    | 0.735             | 0.727             | 0.741             | 0.551             |
> > | LaTo                            |                   |                   |                   |                   |
> > | Cross-id landmark               | 0.768             | 0.741             | 0.762             | 0.580             |
> > | Self-id landmark by LP          | **0.777**         | **0.748**         | **0.775**         | **0.594**         |
> > | Dropout 10% Self-id landmark  | 0.771 (-0.6%)     | 0.738 (-1.0%)     | 0.768 (-0.7%)     | 0.586 (-0.8%)     |
> > | Dropout 30% Self-id  landmark  | 0.753 (-2.4%)     | 0.722 (-2.6%)     | 0.745 (-3.0%)     | 0.569 (-2.5%)     |
> > | Dropout 50% Self-id landmark   | 0.733 (-4.4%)     | 0.697 (-5.1%)     | 0.714 (-6.1%)     | 0.538 (-5.6%)     |
> >
> >
> > **[W]: Cfg is not mentioned in the abstract and the main contribution is not included in the framework.**
> >
> > **[A]:** We have followed the kind suggestion and reorganized the abstract and the framework (Figure 4) in the revised manuscript.
> >
> > [1] Buolamwini, Joy, and Timnit Gebru. "Gender shades: Intersectional accuracy disparities in commercial gender classification." Conference on fairness, accountability and transparency. PMLR, 2018.

---

> > > ### Author Response · Authors · 2025-11-26
> > >
> > > Dear Reviewer,
> > >
> > > We have conducted additional experiments to evaluate the generalization ability of LaTo to full-body editing. Together with our original results, we now demonstrate LaTo’s capability on three tasks: complex face image editing, full-body image editing, and facial animation. The corresponding qualitative results and comparisons are included in our revised version (Figures 6, 7, 11, and 12).
> > >
> > > We hope that our revisions and responses have fully addressed your concerns. As the discussion phase concludes, we would greatly appreciate your further feedback and would be glad to clarify any remaining issues.

---

### Official Review · Reviewer_ApJ8 · 2025-10-27

**Soundness:** 4
**Presentation:** 4
**Contribution:** 4
**Rating:** 6
**Confidence:** 4

**Summary:**

This paper achieves better facial control while maintaining high identity fidelity by tokenizing key points, as opposed to other methods that condition on rendered landmark images. Furthermore, the integration of a large model supporting text input further enriches the application scenarios. Additionally, the authors propose a new face editing dataset. I believe this is a complete and highly practical paper.

**Strengths:**

1. By tokenizing key points, this method achieves better facial control while maintaining high identity fidelity, in contrast to other approaches that condition on rendered landmark images.
2. The support for text input further expands the application scenarios.
3. The data and experiments are sufficient, and the writing is clear and easy to understand.

**Weaknesses:**

1. The DIT structure results in significant resource consumption, and it is necessary to evaluate the inference speed.
2. Keypoints have limited capability in modeling geometric information and are not fully decoupled from identity, which may pose challenges in cross-identity tasks.

**Questions:**

1. In cross-identity scenarios, could there be an issue with inadequate decoupling between geometry and identity?
2. What is the inference speed?

**Details Of Ethics Concerns:**

I don't have many concerns; the authors have already explained it clearly.

---

> ### Author Response · Authors · 2025-11-21
> **Response to Reviewer ApJ8 (Part 1/1)**
>
> **[Q1]: In cross-identity scenarios, could there be an issue with inadequate decoupling between geometry and identity?**
>
> **[A1]:** To assess the effectiveness of cross‑identity landmark conditioning, we collect 20 portraits spanning diverse ages, genders, and ethnicities. For each portrait, we sample 10 cross‑identity landmarks as input. As shown in the table below, LaTo continues to significantly outperform the Step1X‑Edit† (finetuned on HFL-150K) across all metrics about 3%. This robustness primarily arises from three factors. First, the landmark tokenizer disentangles explicit shape-related structure from pixel-level appearance, which stabilizes geometric reasoning and facilitates the capture of generalized expression dynamics and identity-agnostic pose variations under cross-identity conditions. Additional experiments discussed in Reviewer oX4b’s Q5 further show that quantization produces more unified facial representations and improves generalization. Second, the token fuser encourages the model to interpret driving facial geometry, editing instructions, and visual identity features in a coordinated manner. This balances their contributions and enables driving-geometry control in an implicit, feature-level manner, rather than relying on explicit geometric following. Third, the coupling strength between driving geometry and identity can be flexibly adjusted through the learned unconditional landmark tokens. Qualitative cross-identity examples in image editing and video facial animation (Figures 6 and 11) have been included in our paper!
> | Methods       | SC      | VQ      | NA      | IP      |
> |---------------|---------|---------|---------|---------|
> | Step1X-Edit†  | 0.735   | 0.727   | 0.741   | 0.551   |
> | LaTo          | **0.768** | **0.741** | **0.762** | **0.580** |
>
> **[Q2]: What is the inference speed?**
>
> **[A2]:** We evaluate the inference speed of all methods on a single A100 GPU with 1024 px inputs. For advanced editing models, efficiency is largely constrained by their complex architectures. The inference time of LaTo is dominated by the base model Step1X-Edit, while the additional overhead from our token fuser and landmark predictor is negligible, adding only 1.83 s and 0.67 s respectively (with vLLM inference).
>
> | Methods            | Instruct-P2P | AnyEdit | OmniGen | Bagel | Step1X-Edit† | Flux.1-Kontext† | LaTo |
> |--------------------|-------------|---------|---------|-------|--------------|------------------|------|
> | Inference time (s) | 14.4        | 18.7    | 67.3    | 54.8  | 49.6         | 43.2             | 52.1 |

---

### Official Review · Reviewer_oX4b · 2025-10-29

**Soundness:** 3
**Presentation:** 3
**Contribution:** 3
**Rating:** 6
**Confidence:** 4

**Summary:**

This paper propose LaTo, a landmark-tokenized diffusion transformer for fine-grained and identity-preserving face editing, and curate a face editing dataset HFL-150K. The core designs of LaTo are (i) landmark tokenization of raw coordinates, (ii) location-mapping positional encoding, and (iii) landmark-aware classifier-free guidance. Experiments demonstrate that LaTo achieves state-of-the-art performance.

**Strengths:**

- The edited images presented in this work demonstrate effective fine-grained control, with experiments showing improved performance over previous state-of-the-art methods. It enables more precise and adaptable face editing, holding significant practical value for relevant applications.
- The overall pipeline is logical and intuitive, and the writing is well-structured.
- Dataset contribution: This work develops an automated synthesis-and-curation pipeline and constructs a face-editing dataset comprising over 150K face pairs annotated with fine-grained instructions.

**Weaknesses:**

1. Weaknesses in Ablation Studies: Although the paper includes some ablation studies, they are relatively limited. Please provide the rationale for not using traditional landmark extractors as the landmark predictor, along with supporting ablation studies on its utility. Furthermore, it would be beneficial to include ablation analyses on the codebook size and the design of the multi-modal token fuser.
2. Lack of user study. For the fine-grained face editing task, the quantitative objective metrics are insufficient to fully validate the effectiveness of the editing results.

**Questions:**

1. Why did you choose to use a VLM for landmark prediction instead of traditional, well-established landmark detectors? Were any experiments conducted to demonstrate the effectiveness of your chosen approach?
2. Was the accuracy of the head rotation estimation evaluated qualitatively? (Alternatively, if emphasizing the lack of such evaluation: The experimental evaluation appears incomplete. Specifically, is there any qualitative assessment of the accuracy for the estimated head rotation angles?
3. Was a user study conducted?
4. Were there any ablation studies on the proposed token fusing mechanism? How does it compare to other fusion strategies?
5. How was the size of the codebook determined? Were ablation studies performed to validate the chosen size?

---

> ### Author Response · Authors · 2025-11-21
> **Response to Reviewer oX4b (Part 1/2)**
>
> **[Q1]: Why did you choose to use a VLM for landmark prediction instead of traditional, well-established landmark detectors? Were any experiments conducted to demonstrate the effectiveness of your chosen approach?**
>
> **[A1]:** Although using standard landmark detectors as editing conditions may seem straightforward, it creates a chicken and egg dilemma. At inference the target image is unavailable, so landmark locations are unknown and cannot be reliably extracted, yet without landmarks the editing pipeline cannot proceed. We address this with a VLM that predicts plausible, identity preserving facial landmarks by leveraging visual causal reasoning and instruction alignment, combined with CoT  and supervised fine tuning. We evaluate two alternatives on HFL-150K: (1) generate with Step1X-Edit† and then extract landmarks using landmark detectors, and (2) train a Transformer following [1] on the same data as landmark predictor. The results show that landmarks extracted from Step1X-Edit† further improve IP, which indicates LaTo learns more generalized facial geometry features for identity-preserving control. However, route (1) requires generating the target image first, which increases inference time and is restricted by the performance of the editing model. The Transformer baseline yields less accurate landmarks due to weaker representation learning than the VLM. Our LP achieves the best overall performance under vLLM inference (0.67s).
> | Methods                 | SC    | VQ    | NA    | IP    | Latency(s) |
> |-|-|-|-|-|-|
> | Step1X-Edit†            | 0.804 | 0.725 | 0.801 | 0.571 | **49.6**      |
> | Step1X-Edit† + LaTo     | 0.825 | 0.731 | 0.811 | 0.615 | 101.7     |
> | Transformer + LaTo      | 0.818 | 0.727 | 0.805 | 0.608 | 51.4      |
> | Our LP + LaTo           | **0.832** | **0.749** | **0.805** | **0.634** | 52.1      |
>
> **[Q2]: Was the accuracy of the head rotation estimation evaluated qualitatively? Specifically, is there any qualitative assessment of the accuracy for the estimated head rotation angles?**
>
> **[A2]:** We selected 433 head-pose editing cases from HFL-150K. Accuracy was evaluated using the pose discriminator (Lines 202–208) and human ratings from 21 participants. A sample is correct if its angular deviation from the instructed pose is within 10°, and scale lines are overlaid to aid human judgment. As shown in the table below, about 80% of cases follow the instructed pose, and human judgments closely match the discriminator.
> | Metrics   | pose discriminator | human evaluation |
> |-|-|-|
> | Accuracy | 0.836              | 0.806            |
>
> **[Q3]: Was a user study conducted?**
>
> **[A3]**: We conducted a user study on 1,110 cases across three benchmarks, evaluating semantic consistency, visual quality, and identity preservation on a 5-point Likert scale. Detailed questionnaires are provided in Lines 1317-1342. For fair comparison, we compare Flux.1-Kontext† and Step1X-Edit†, both fine-tuned on our dataset, and collect 21 valid responses. As shown in the table below, LaTo’s overall rating exceeds Flux.1-Kontext† by about 4.1%, demonstrating superior fine-grained editing performance.
> | Methods        | SC      | VQ      | IP      | Overall |
> |-|-|-|-|-|
> | Step1X-Edit†   | 0.728   | 0.810   | 0.746   | 0.761   |
> | Flux.1-Kontext†| 0.712   | 0.824   | 0.774   | 0.770   |
> | LaTo           | **0.766** | **0.837** | **0.829** | **0.811** |

---

> > ### Author Response · Authors · 2025-11-21
> > **Response to Reviewer oX4b (Part 2/2)**
> >
> > **[Q4]: Were there any ablation studies on the proposed token fusing mechanism? How does it compare to other fusion strategies?**
> >
> > **[A4]:** We evaluated two mainstream fusion strategies for injecting landmark tokens: (1) feature addition via a ControlNet block an (2) additional cross attention layer. Table below reports the results on HFL-150K. Feature addition has two drawbacks. First, sequence length mismatch between landmark and image tokens breaks spatial correspondence and reduces flexibility. Second, it restricts interactions between landmark tokens and instruction features. The added cross attention removes spatial constraints but weakens coordination between instruction and landmarks, producing unstable training where the model tend to overfit driving-geometry. Our fuser enables flexible interactions among misaligned tokens and encourages joint conditioning during training. Attention visualizations also show that, with identical landmarks, changing the instruction modulates landmark-to-image attention, confirming effective coordination.
> >
> > | Fusion strategies | SC      | VQ      | NA      | IP      |
> > |-------------------|---------|---------|---------|---------|
> > | Feature addition  | 0.793   | 0.725   | 0.774   | 0.596   |
> > | Cross-attention   | 0.812   | 0.731   | 0.792   | 0.605   |
> > | Our fuser         | **0.832** | **0.749** | **0.805** | **0.634** |
> >
> > **[Q5]: How was the size of the codebook determined? Were ablation studies performed to validate the chosen size?**
> >
> > **[A5]:** We evaluated codebook sizes of 1,024, 8,192, and 16,384, as well as a no‑quantization ablation. Table below reports landmark reconstruction performance on HFL‑150K. For PSNR, we render the landmarks into images. The L1 offset is evaluated at 256 px resolution. It indicates that removing quantization forces the tokenizer to operate on continuous and less consistent facial features, which degrades performance. This confirms that discrete and general landmark tokens provide more stable geometric representations and improve generalization. Although larger codebooks offer higher nominal capacity, the 16,384‑entry codebook resulted in unstable optimization and significant codebook underutilization, which reduced its effective capacity. The 8,192‑entry codebook achieved the best overall performance. We anticipate further gains through improved codebook utilization.
> >
> > | Codebook Size  | Utilization (%↑) | L1 distance(↓) | PSNR(↑)  |
> > |----------------|------------------|----------------|----------|
> > | no‑quantization| -                | 7.04           | 19.11    |
> > | 1024           | **81.42**        | 5.32           | 20.47    |
> > | 8192           | 70.39            | **3.67**       | **21.33**|
> > | 16384          | 44.75            | 4.51           | 20.61    |
> >
> > [1] Identity-Preserving Talking Face Generation with Landmark and Appearance Priors, CVPR23.

---

### Official Review · Reviewer_Cy7B · 2025-11-01

**Soundness:** 3
**Presentation:** 3
**Contribution:** 2
**Rating:** 4
**Confidence:** 3

**Summary:**

This paper introduces LaTo, a landmark-tokenized diffusion transformer for fine-grained, identity-preserving human face editing.
LaTo quantizes landmark coordinates into discrete facial tokens, integrating them into diffusion transformers using location-mapped positional encoding.
It also introduces a landmark predictor based on a fine-tuned vision–language model with structured chain-of-thought reasoning, allowing intuitive user control without explicit landmark input.
The authors further curate HFL-150K, a large-scale dataset containing 150k real and synthetic face editing pairs with precise annotations. Extensive experiments demonstrate that LaTo achieves superior identity preservation and semantic consistency compared to prior state-of-the-art models.

**Strengths:**

1. Direct landmark tokenization provides an efficient geometric prior for diffusion transformers.
2. LaTo achieves consistent improvement across benchmarks (HFL-150K, ICE-Bench, GEdit-Bench).
3. The effect of positional encoding and classifier-free guidance is well quantified.
4. HFL-150K is a valuable, large-scale dataset that provides both real and synthetic samples with fine-grained instructions — a useful resource for the community.

**Weaknesses:**

1. The landmark predictor tends to produce nearly identical landmark configurations for the same expression instruction (e.g., “make him happy strongly”) across subjects, for example, the first four columns of Figure 1.
This over-templating reduces individual expressiveness and may lead to uniform, less personalized facial deformations.
The paper lacks analysis on inter-identity landmark variance.
2. Figure 1 (second to last column) includes a cartoon-style input image as an example. In such cases, landmark prediction becomes substantially more difficult because the facial geometry deviates from human anatomy — e.g., mouths drawn as single lines, exaggerated eyes, missing noses, or highly stylized jawlines.
These features make standard 68-point facial landmark detection ill-defined.
However, the paper does not clarify whether the proposed landmark predictor is trained or evaluated on such stylized data.
Given that HFL-150K is described as primarily real-human-face-based (Section 3.1), it is doubtful that the model has seen sufficient cartoon or synthetic faces during training.
3. Although the paper promises public release “upon acceptance,” both LaTo’s code and the HFL-150K dataset are currently unavailable.
For a paper emphasizing both model and dataset contributions, this limits reproducibility.

**Questions:**

1. How are identity-dependent facial variations (e.g., different muscle or bone structures) handled when generating landmarks for the same expression?
2. How are identity preservation metrics validated against human perceptual ratings? Is there a correlation analysis?
3. What is the sensitivity of the model to noisy or inaccurate landmark predictions from the VLM?
4. Can the proposed landmark predictor generate accurate and stable landmark coordinates for non-realistic, stylized cartoon faces? Does the full LaTo pipeline (landmark tokenization + DiT fusion) maintain editing quality when the input lies outside the natural-face manifold?
5. Can the landmark tokenizer or location-mapping positional encoding be applied to other structured domains, such as full-body or hand motion editing?
6. How does the landmark predictor perform under noisy or ambiguous instructions?
7. Are there any plans to release HFL-150K in subsets (e.g., synthetic only)?

---

> ### Author Response · Authors · 2025-11-21
> **Response to Reviewer Cy7B (Part 1/2)**
>
> **[Q1]: How are identity-dependent facial variations handled when generating landmarks for the same expression?**
>
> **[A1]:** We appreciate the concern and clarify that the first four columns of Figure 1 deliberately use a single exaggerated landmark template, independent of any input face, to show how rendered-image baselines can overfit a fixed template. We have revised the caption accordingly. To assess inter-identity landmark variance, we evaluate LP on 10 class instructions, each applied to 20 portraits with diverse ages, genders, and ethnicities, and measure variation by (1) pairwise normalized landmark L1 distance and (2) ArcFace similarity between LaTo-edited images. As reported in the table below, we observe clear identity-dependent landmark variations under identical instructions, with the average deviation of a corresponding landmark point being about 15% of the face area. Since our CoT explicitly analyzes and encodes facial information in the first step, it steers the VLM away from uniform landmarks and toward self-identity-driven configurations.
>
> | Instructions | Happy | Angry | Disgusted | Sad   | Surprised | Scared | Left 30° | Left 60° | Right 30° | Right 60° |
> |-|-|-|-|-|-|-|-|-|-|-|
> | L1 loss     | 0.142 | 0.155 | 0.162     | 0.148 | 0.135     | 0.158  | 0.168    | 0.182    | 0.171     | 0.185     |
> | ArcFace     | 0.034 | 0.029 | 0.027     | 0.032 | 0.039     | 0.028  | 0.026    | 0.022    | 0.027     | 0.023     |
>
> **[Q2]: How are identity preservation metrics validated against human perceptual ratings? Is there a correlation analysis?**
>
> **[A2]:** We perform a cross-validation analysis comparing automatic metrics (CLIPScore, FID, and IP score) against the user study, with experimental details provided in Q3 for Reviewer oX4b. For each sample, we identify the metric whose ranking best matches human assessments of identity consistency and report agreement rates. Results show that CLIPScore has only moderate alignment with human perception, FID is inversely correlated, and our IP score shows strong concordance with human judgments. We acknowledge optimal identity preservation requires balancing subject consistency and instruction fidelity, and view refining the IP metric to better capture this trade-off as an important direction for future work.
>
> | Metrics       | CLIPScore | FID   | IP Score |
> |-|-|-|-|
> | Alignment  | 59.4%     | 21.7% | 85.8%    |
>
> **[Q3]: What is the sensitivity of the model to noisy or inaccurate landmark predictions from the VLM?**
>
> **[A3]:** We define noisy or inaccurate landmarks as those with (1) incomplete landmarks or (2) cross-identity semantics. For the 20 portraits in Q1, we use 10 cross-identity landmarks paired with instructions from HFL150K and 10 landmarks generated by LP under the same instructions. Noise is simulated by randomly dropping landmark points and replacing them with unconditional tokens. As shown in the table below, both cross-identity inputs and minor dropout cause only negligible degradation in identity preservation, while still significantly outperforming the finetuned base model. This robustness mainly comes from three components: (1) the landmark tokenizer disentangles explicit shape-related features from pixel-level appearance, facilitating the capture of generalized expression dynamics and identity-agnostic pose variations under noisy or cross-identity inputs, and its quantization yields more generalizable facial representations (see our Q5 response to Reviewer oX4b); (2) the token fuser enables decoupled interactions among driving geometry, identity, and text, providing implicit, feature-level geometry control and mitigating overfitting to imperfect landmark conditions; (3) the learned unconditional landmark tokens give each landmark position a flexible way to adjust the coupling between geometry and identity, so that even when landmark absence becomes severe (50%), LaTo, although showing noticeable degradation in identity preservation, still maintains basic visual quality and key facial traits. The visualizations are provided in Figure 6.
>
> | Methods                         | SC                | VQ                | NA                | IP                |
> |-|-|-|-|-|
> | Step1X-Edit†                    | 0.735             | 0.727             | 0.741             | 0.551             |
> | LaTo                            |                   |                   |                   |                   |
> | Cross-id landmark               | 0.768             | 0.741             | 0.762             | 0.580             |
> | Self-id landmark by LP          | **0.777**         | **0.748**         | **0.775**         | **0.594**         |
> | Dropout 10% Self-id   | 0.771 (-0.6%)     | 0.738 (-1.0%)     | 0.768 (-0.7%)     | 0.586 (-0.8%)     |
> | Dropout 30% Self-id    | 0.753 (-2.4%)     | 0.722 (-2.6%)     | 0.745 (-3.0%)     | 0.569 (-2.5%)     |
> | Dropout 50% Self-id   | 0.733 (-4.4%)     | 0.697 (-5.1%)     | 0.714 (-6.1%)     | 0.538 (-5.6%)     |

---

> > ### Author Response · Authors · 2025-11-21
> > **Response to Reviewer Cy7B (Part 2/2)**
> >
> > **[Q4]: Can the proposed landmark predictor generate accurate and stable landmark coordinates for non-realistic, stylized cartoon faces? Does the full LaTo pipeline maintain editing quality when the input lies outside the natural-face manifold?**
> >
> > **[A4]:** Our LP is trained primarily on realistic portraits, which limits its ability to infer accurate landmarks for non-realistic or stylized cartoon faces, where standard 68-point landmarks are often difficult to extract. Nevertheless, the full LaTo pipeline maintains strong editing quality on cartoon, AIGC, and oil painting styles (Figure 6). We attribute this to the pipeline’s robustness to noisy landmark inputs, as demonstrated in Q3, and to the strong generalization capability of the underlying base model.
> >
> > **[Q5]: Can the landmark tokenizer or location-mapping positional encoding be applied to other structured domains, such as full-body or hand motion editing?**
> >
> > **[A5]:** Yes! In our follow-up work, we extend both designs to facial animation. We build a 16k high-quality portrait video dataset with similar filtering criteria. The tokenizer is upgraded to 3D with an 81-frame temporal window. For fusion, we use Wan2.2-TI2V-5B as the base model and prepend landmark tokens to video tokens for unified representation. As shown in Figure 11, the extended LaTo preserves identity well when editing with cross-identity reference face videos, indicating strong scalability. For full-body cases, joint keypoints offer physical signals analogous to facial landmarks, so we expect similarly disentangled geometric control under large pose differences. We are collecting full-body data for training and will report results once available.
> >
> > **[Q6]: How does the landmark predictor perform under noisy or ambiguous instructions?**
> >
> > **[A6]:** We evaluate three types of ambiguous or noisy conditions: composite expressions (e.g., happy and surprised), conflicting pose instructions (e.g., turn left and turn right), and extreme facial poses. For each, we construct 120 cases and assess quality manually for landmark accuracy and automatically with Qwen3-VL for geometric similarity and semantic consistency. As shown in the table below, LP is robust to instruction-level ambiguity, benefiting from strong language–vision alignment and structured geometric reasoning. For composite expressions, it generates anatomically plausible intermediates, likely by interpolating in the landmark space learned from diverse single-expression data. For conflicting pose instructions, the CoT module infers a coherent orientation and often defaults to a near-frontal pose when cues cancel out, which avoids erratic point placements. Image-level noise has a stronger impact: extreme poses suffer from heavy occlusion and are underrepresented in the training set. We plan to incorporate 3D-aware priors to alleviate the extreme pose limitation. Quantitative results are given in Figure 7.
> >
> > | Ambiguity type                     | geometric similarity | semantic consistency | Accuracy |
> > |------------------------------------|----------------------|----------------------|----------|
> > | composite expressions instructions | 0.681                | 0.584                | 0.608    |
> > | conflicting pose instructions      | 0.743                | 0.601                | 0.642    |
> > | extreme facial poses               | 0.502                | 0.483                | 0.417    |
> >
> > **[Q7]: Are there any plans to release HFL-150K in subsets (e.g., synthetic only)?**
> >
> > **[A7]:** Yes! the LaTo code has been uploaded to the anonymous repository. As stated, we will release the full HFL150K dataset upon acceptance. Thank you for your attention!

---

> > > ### Author Response · Authors · 2025-11-26
> > > **Response to Reviewer Cy7B (Part 3)**
> > >
> > > **[Q5]: Additional experimental results of Q5**
> > >
> > > **[A5]:** We have applied the landmark tokenizer and location-mapping positional encoding to full-body editing. Specifically, we collect 24.3k full-body image pairs from TikTok [1], OpenHumanVid, and Champ [2]. We compute the foreground optical flow and retain only samples with sufficient body motion. To quickly verify LaTo’s design, we remove instruction annotations and use only OpenPose as the control signal, turning this into a pure pose-driven body editing task. We train both the pose–image baseline and LaTo on 24.2k pairs, and use the remaining 100 self-identity samples plus 50 cross-identity poses for evaluation. We evaluate visual quality (VQ)and subject consistency (SC) with Qwen2.5-VL, and the results are reported below. We observe that cross-identity input poses a greater challenge for full-body editing than for faces, as body shapes exhibit substantially higher inter-subject variability. The baseline’s pixel-wise alignment always “pastes” the body shape from the conditional image, which severely distorts identity. In contrast, LaTo preserves the input person’s overall body structure while still following the target pose condition, leading to more stable and identity-consistent body editing. A detailed explanation is provided in our response to Q3, and related qualitative comparisons are included in the Figure 12.
> > >
> > > | Methods             | VQ(Self-identity) | SC(Self-identity) |  VQ(Cross-identity) |  SC(Cross-identity) |
> > > |---------------------|----------------------------|----------------------------|-----------------------------|-----------------------------|
> > > | Body-image baseline | 0.713                       | 0.604                       | 0.615                        | 0.486                        |
> > > | LaTo                | **0.739**                       | **0.656**                       | **0.697**                        | **0.598**                       |
> > >
> > > [1]Learning High Fidelity Depths of Dressed Humans by Watching Social Media Dance Videos, CVPR21.
> > >
> > > [2]Champ: Controllable and Consistent Human Image Animation with 3D Parametric Guidance, ECCV24.
> > >
> > >
> > >
> > > Dear Reviewer Cy7B,
> > > We hope that our responses have thoroughly addressed your questions. As the discussion phase comes to a close, we eagerly await your feedback and would appreciate knowing if you have any remaining concerns we can address. We would be grateful if you find our revisions satisfactory and consider raising your evaluation of our paper.

---

### Author Response · Authors · 2025-11-21

We thank all reviewers for their valuable suggestions. In the revised version, we (1) add qualitative editing results under challenging landmark conditions, including cross-identity and occlusion, for diverse real and stylized portraits (ApJ8, Cy7B, bTdN); (2) provide visualizations of extending LaTo to video-based facial animation (bTdN, Cy7B); (3) report quantitative results of the landmark predictor under realistic noisy conditions (Cy7B); (4) include user study results and the detailed questionnaire (oX4b); (5) present diversity statistics of HFL-150K (bTdN); (6) report inference time comparisons (ApJ8); and (7) add an ablation study on the codebook size (oX4b). We also revise the main paper by adding a description of the landmark-aware CFG to the abstract and listing the main contributions in the framework figure (bTdN). Additional quantitative results are included in our responses to each reviewer and will be incorporated into the paper to further strengthen our contributions. The source code of LaTo has also been uploaded at (https://anonymous.4open.science/r/16449f5f-a175-4146-8cc1-6935fb7aea65). We eagerly await your feedback and would appreciate knowing if you have any remaining concerns we can address.

---

### Author Response · Authors · 2025-12-02
**Summary of Rebuttal by Authors (part 1/2)**

Dear Handling Area Chair,

We would like to extend our heartfelt gratitude for the time and effort devoted to the review process and for the pertinent questions raised. We are deeply encouraged by the reviewers’ praise and commendations. Below we directly quote some of their original comments:

**[Acknowledged Novelty] (Reviewers Cy7B, bTdN, and ApJ8)**

Reviewers consistently emphasize that “direct landmark tokenization provides an efficient geometric prior for diffusion transformers” (Cy7B) and is an “innovative approach that moves beyond traditional pixel-based conditioning in face editing tasks” (bTdN), “achieving better facial control while maintaining high identity fidelity” (ApJ8) and helping “prevent identity drift when landmarks differ significantly from the source image” (bTdN), with the effects of positional encoding and classifier-free guidance “well quantified” (Cy7B).

**[Consistent Performance & Comprehensive Evaluation] (Reviewers Cy7B, oX4b, ApJ8, and bTdN)**

LaTo “achieves consistent improvement across benchmarks (HFL-150K, ICE-Bench, GEdit-Bench)” (Cy7B), with “experiments showing improved performance over previous state-of-the-art methods” (oX4b). Reviewers note that “the data and experiments are sufficient” (ApJ8) and that “comprehensive ablation studies and both qualitative and quantitative evaluations demonstrate LaTo’s effectiveness across multiple metrics, supporting its proposed innovations” (bTdN).

**[Valuable Dataset & Data Pipeline](Reviewers Cy7B, oX4b, and bTdN)**

Reviewers underline that “HFL-150K is a valuable, large-scale dataset that provides both real and synthetic samples with fine-grained instructions — a useful resource for the community” (Cy7B), and that it is “a valuable contribution that is essential for training advanced face editing models” (bTdN). The work also “develops an automated synthesis-and-curation pipeline and constructs a face-editing dataset comprising over 150K face pairs annotated with fine-grained instructions” (oX4b).

**[Fine-Grained & Practical Face Editing] (Reviewers oX4b and ApJ8)**

The edited images “demonstrate effective fine-grained control”, and LaTo “enables more precise and adaptable face editing, holding significant practical value for relevant applications” (oX4b). In addition, “the support for text input further expands the application scenarios” (ApJ8). Reviewers also find that “the overall pipeline is logical and intuitive, and the writing is well-structured” (oX4b) and “clear and easy to understand” (ApJ8), making the approach easy to use in practice.

---

> ### Author Response · Authors · 2025-12-02
> **Summary of Rebuttal by Authors (part 2/2)**
>
> During the response phase, we directly addressed the concerns and clarified misunderstandings raised by the reviewers:
>
>   - **[Robustness to cross-identity and inaccurate landmarks] (Reviewer Cy7B Q3, Reviewer ApJ8 Q1, Reviewer bTdN Q3)**
>     We evaluated LaTo with cross-identity landmarks exhibiting large differences and with randomly masked self-identity landmarks. LaTo still outperforms the baseline (Step1X‑Edit†, finetuned on HFL-150K) by about 3% across all metrics under cross-identity inputs. When fewer than 30% of landmarks are inaccurate, identity preservation drops by only 0.8%–2.5%. Even with 50% inaccurate landmarks, LaTo maintains basic visual quality and key facial traits. We also added qualitative examples in Figure 6 and a robustness analysis in Lines 537–546.
>
>   - **[Human perceptual evaluation] (Reviewer oX4b Q2 & Q3, Reviewer Cy7B Q2)**
>     We conducted several human studies: (1) a 5-point Likert user study on fine-grained editing, where LaTo’s overall rating exceeds Flux.1-Kontext by 4.1%; (2) a head-rotation evaluation using both human raters and an expert model, with LaTo’s head-editing accuracy around 80% under both assessments; (3) a cross-validation analysis comparing automatic metrics against human ratings for identity preservation. Our IP score shows strong alignment with human judgments (85.8%), clearly higher than CLIPScore (59.4%) and FID (21.7%). Detailed questionnaires are provided in Lines 1353–1378.
>
>   - **[Generalization to other domains] (Reviewer Cy7B Q5, Reviewer bTdN Q1)**
>     We extended LaTo to facial video animation and full-body image editing, where it still demonstrates stronger identity preservation than rendered-image conditioning. We provided experimental details in Lines 942–1034 and qualitative results in Figures 11 and 12.
>
>   - **[Diverse realistic and non-realistic inputs] (Reviewer Cy7B Q4, Reviewer bTdN Q2)**
>     We evaluated LaTo on diverse portraits across different ages, ethnicities, genders, and styles (cartoon, oil painting, AGI-generated). Figure 6 shows robust performance, supported by the diversity of HFL‑150K and the generalizable facial representations enabled by landmark tokenization. We also added demographic statistics of HFL‑150K in Lines 891–915.
>
>   - **[Inference time] (Reviewer ApJ8 Q2, Reviewer bTdN Q1)**
>     Our designs increase the baseline inference time only slightly, from 49.6s to 52.1s. With several acceleration strategies, we further reduce the runtime to 4.9s per image without noticeable degradation in output quality.
>
>   - **[Ablations on token fusing and codebook size] (Reviewer oX4b Q4 & Q5)**
>     We compared our token fusing mechanism against feature addition and additional cross-attention. Our design improves semantic consistency and identity preservation by about 2.0% and 2.9%, respectively, over the second-best variant. For codebook size, the chosen setting yields the best reconstruction quality (PSNR higher by 0.72 than the second-best variant), with codebook utilization above 70%. We provide a more detailed performance analysis in Lines 1221–1227.
>
>   - **[Identity-dependent landmark variations of the same instruction] (Reviewer Cy7B Q1)**
>     We clarified the misunderstanding regarding the four columns in Figure 1 and evaluated the landmark predictor with 10 instruction types. The predicted results show that the average deviation of corresponding landmark points is about 15% of the face area, confirming that our landmark predictor does not lead to uniform, less personalized facial deformations.
>
>   - **[The rationale of landmark predictor] (Reviewer oX4b Q1)**
>     We clarified that traditional landmark extractors cannot provide edited self-identity landmarks out of thin air, and we compared our landmark predictor with two variants, validating its effectiveness and necessity.
>
>   - **[Ambiguous instructions for the landmark predictor] (Reviewer Cy7B Q6)**
>     We evaluated composite expressions and conflicting pose instructions. The accuracy under such instruction-level ambiguity reaches 60.8% and 64.2%, respectively, demonstrating the robustness of the landmark predictor. Quantitative results are provided in Figure 7.
>
>   - **[Reproducibility] (Reviewer Cy7B Q7)**
>     We provided the full LaTo code and a subset of HFL‑150K via anonymous links, and we will release the full HFL‑150K dataset publicly upon acceptance.
>
> Thank you very much for your time and consideration.
>
> Best,
>
> LaTo Author(s) Team

---

### Meta-Review · Area_Chair_CRyU · 2026-01-06

**Summary:**

The paper proposes LaTo, a landmark-tokenized diffusion transformer for fine-grained, identity-preserving face editing. The paper also proposes a  large-scale dataset, named  HFL-150K. The reviewers generally found the core idea technically sound and practically valuable. The main concerns centered on (1) robustness to cross-identity/inaccurate landmarks and OOD inputs (stylized faces, extreme poses/occlusion), (2) evaluation completeness (user study, ablations), (iv) inference efficiency. The authors have partially adressed the concerns. Overall, the AC considers this to be a solid and acceptable ICLR paper.

**Reviewer Concerns:**

Some of the concerns are addressed, which are listed as follows.
1. Clarified the Figure 1 misunderstanding and provided quantitative analysis showing identity-dependent landmark variance under the same instruction. 2. Added robustness evaluations for cross-identity landmarks and landmark noise. 3. Added human perceptual studies. 4. Added ablations on token fusion strategies, codebook size, and landmark predictor alternatives. 5. Reported inference-time comparisons. 6.  Provided additional evidence for generalization to video facial animation and full-body editing.

Some of the outstanding concerns are as follows.
1. Landmark predictor accuracy on heavily stylized/cartoon faces are not fully resolved. The rebuttal also acknowledges these limitations.

**Reviewer Scores:**

Reveiwer Cy7B: likely maintains or increases score given clarified misunderstanding, added robustness tests, and release commitments.
Reveiwer oX4b: likely increases acore after added user study and stronger ablations.
Reveiwer ApJ8: likely maintains acores as cross-identity robustness and inference speed are addressed.
Reveiwer bTdN: likely maintains or slightly increases scores given added diversity stats, broader-domain generalization, robustness experiments.

---

### Decision · Program_Chairs · 2026-01-26

Accept (Poster)